# MECHANISTIC EVALUATION OF TRANSFORMERS AND STATE SPACE MODELS

## ABSTRACT

State space models (SSMs) for language modelling promise an efficient and performant alternative to quadratic-attention Transformers, yet show variable performance on recalling basic information from the context. While performance on synthetic tasks like Associative Recall (AR) can point to this deficiency, behavioural metrics provide little information as to *why*—on a mechanistic level—certain architectures fail and others succeed. To address this, we conduct experiments on AR, and find that only Transformers and Based SSM models fully succeed at AR, with Mamba and DeltaNet close behind, while the other SSMs (H3, Hyena) fail. We then use causal interventions to explain why. We find that Transformers and Based learn to store key–value associations in-context using induction. By contrast, the SSMs seem to compute these associations only at the last state using a single layer. We further investigate the mechanism underlying the success of Mamba, and find novel evidence that Mamba *does* implement induction: not via the SSM, but instead via short convolutions. Further experiments on a new hierarchical retrieval task, Associative Treecall (ATR), show that all architectures learn the same mechanism as they did for AR. Furthermore, we show that Mamba can learn Attention-like induction on ATR when short convolutions are removed. These results reveal that architectures with similar accuracy may still have substantive differences, motivating the adoption of mechanistic evaluations.

⌂ anonymous.4open.science/r/tinylang-1061

## 1 INTRODUCTION

Transformers with quadratic attention remain the dominant architecture in language modelling despite numerous proposed efficient alternatives. Most notably, **state space models** (SSMs) achieve impressive perplexities and benchmark scores (e.g. Gu & Dao, 2024). Yet, SSMs exhibit deficiencies that benchmarks often fail to capture; for example, they struggle to perform **retrieval**, i.e. copying from the context (Jelassi et al., 2024; Wen et al., 2024; Waleffe et al., 2024; Bick et al., 2025).

Controlled synthetic tasks can make these limitations clear by isolating specific capabilities and enabling expressive experimentation at small scales across architectures. Particularly, much work has used the **associative recall** (AR) task as a testbed for studying in-context retrieval across architectures. In turn, AR has informed the design of novel LM architectures (e.g. Based; Arora et al., 2024b).

However, in this prior work, performance on synthetic tasks is measured solely via behavioural metrics like task accuracy. This is a missed opportunity: an advantage of these tasks is that they are designed to isolate a *specific behaviour* that implicates a mechanistic solution. For example, LMs should solve AR by storing *key–value* associations in-context at the *value*. For Transformer-based LMs (Vaswani et al., 2017), the mechanism is generally thought to be the **induction head**. (Olsson et al., 2022; Fu et al., 2023). We should therefore directly check whether each architecture learns an induction mechanism, as a way of checking that it has learned a robust solution to the task.

Here, we propose using tools from mechanistic interpretability to directly analyse the mechanisms used to solve AR tasks. We use **causal interventions** (Geiger et al., 2024) on model internals to understand how these tasks are learned and implemented across a variety of architectures (§3). This

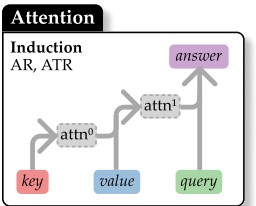 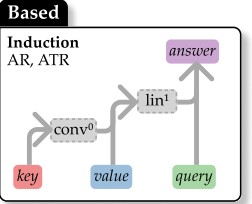 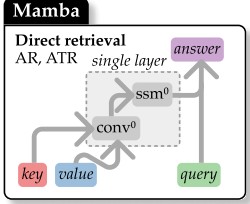 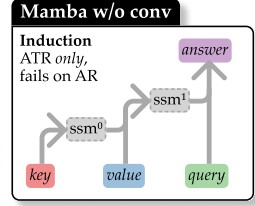

Figure 1: **Associative mechanisms in Attention, Based, and Mamba**: Key results from applying mechanistic metrics to understand how Attention and SSMs solve AR and our new retrieval task, Associative Treecall (ATR). Attention and Based both implement induction, but with different architectural components (§4.2 and §5.2). Mamba instead uses a single layer for both association and retrieval, but uses short convolution for association just like Based (§4.3 and §4.4). When convolutions are removed, Mamba implements induction, but only on ATR (§5.3).

allows us to track the emergence (or lack thereof) of the correct association and retrieval mechanisms inside the model, which moves us beyond observed task accuracy and towards a detailed picture of the solutions different architectures learn.

Through comprehensive experiments on AR, we find that all SSMs except Based do not use induction in the traditional sense to implement AR (§4.2); instead, we find that the best-performing SSM, Mamba, relies heavily on the short convolution component in each layer to perform *key*-*value* association, and fails to learn AR at all without this (§4.3). Through layer-internal interventions, we discover that the short convolution component implements an induction-like associative mechanism in Mamba, with the SSM component only being used for retrieval (§4.4).

To deepen our findings, we introduce **Associative Treecall** (ATR), a novel retrieval task more similar to real-world natural language retrieval than AR (§5). ATR uses a probabilistic context-free grammar (PCFG) to generate hierarchical data, on which we ask AR-like queries; the key difference from AR is that *key*s and their associated *value*s need not be adjacent to each other. We find that the same mechanisms are implicated across architectures on ATR as on AR, confirming the generality of our findings (§5.2). However, unlike AR, Mamba does not need short convolutions to learn ATR; instead, mechanistic metrics reveal that Mamba falls back to an Attention-like two-layer induction mechanism when its short convolutions are removed (§5.3).

We offer a framework for better understanding and evaluating task performance via mechanistic interpretability. We demonstrate multiple examples of mechanistic evaluations revealing fundamental differences between architectures that are unknown via behavioural performance, summarised in Figure 1. We thus introduce mechanistic metrics as a new tool for architecture analysis.

## 2 RELATED WORK

**Associative Recall.** Our work relates to a variety of extant synthetic retrieval tasks. The foremost example, Associative Recall (AR),[1] is a synthetic task that evaluates in-context retrieval for language model architectures. AR has been used extensively, from early work on recurrent neural networks (Graves et al., 2014; Ba et al., 2016; Danihelka et al., 2016; Zhang & Zhou, 2017) to modern SSMs (Fu et al., 2023; Poli et al., 2023; Lutati et al., 2023; Jelassi et al., 2024; Arora et al., 2024a;b; Gu & Dao, 2024; Dao & Gu, 2024; Trockman et al., 2024; Liu et al., 2024a; Okpekpe & Orvieto, 2025; Li et al., 2025b; Wang et al., 2025).

An AR task consists of a sequence of key–value pairs followed by a single *query* key; the goal is to produce the corresponding value for the given query. For example,

$$(1) \qquad \texttt{A 2 C 3 F 9 D 1 C} \rightarrow \textbf{3}$$

Here, the correct next token is 3, since it is the value associated with the key C in context.

---

[1]Also known as *associative retrieval*, *associative memory*, or *induction*.

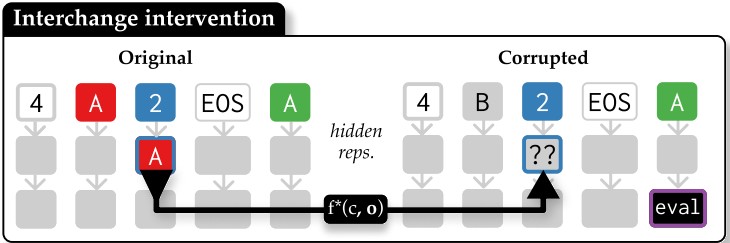

Figure 2: An example interhcange intervention where we corrupt the *key* (A) and attempt to restore it by intervening at the *value* token in an intermediate representation. We evaluate the downstream effect on the next-token prediction at the *query* (which should ideally predict the *answer*).

Despite being synthetic, AR has a direct analogue in natural language: *induction*, referring to in-context copying of sequences (Elhage et al., 2021; Olsson et al., 2022). Arora et al. (2024a;b) show that architecture-level improvements on AR translate directly to induction. Association is additionally widely studied in cognitive science as *binding*. Binding in neural networks has been examined by Greff et al. (2020); Kim & Schuster (2023); Feng & Steinhardt (2024); Prakash et al. (2024); Li et al. (2025a); Prakash et al. (2025).

**Mechanistic interpretability.** In order to measure the contribution of individual model components (neurons, layers, etc.) to output behaviour, we can apply causal interventions on neural network internals (Geiger et al., 2021; 2024). Informally, the core idea is to overwrite an activation at a specific component using a counterfactual input. If this changes model behaviour, then that component is causally relevant to the mechanism underlying that behaviour.

Some prior work in mechanistic interpretability has studied how some language models solve in-context retrieval tasks like induction and multiple choice question answering (Olsson et al., 2022; Lieberum et al., 2023; Brinkmann et al., 2024; Wiegreffe et al., 2025; Bick et al., 2025), as well as the training dynamics of Transformers on toy tasks using mechanistic metrics (Nanda et al., 2023; Reddy, 2024; Singh et al., 2024; Edelman et al., 2024; Tigges et al., 2024; Yin & Steinhardt, 2025). Yet thus far, *architectural comparisons* on synthetic tasks have not made use of causal interventions.

## 3  MECHANISTIC METRICS

Behavioural metrics provide little information as to *why* certain architectures succeed or fail on tasks of interest. Mechanistic metrics, which directly measure how information flows across model components and token positions, can tell us how AR and similar tasks are being solved by different architectures, and thus help us understand failures. We illustrate our approach in Figure 2.

We use interchange interventions (Geiger et al., 2021; 2024) to understand and measure how solutions to AR are implemented across architectures. We introduce this operation and define the resulting metrics for our tasks below. Our implementation uses the pyvene library (Wu et al., 2024).

**Interchange intervention.** Consider a language model $p(\cdot)$ and some input $\mathbf{b}$. We select a component $f$ inside that model which computes some internal representation $f(\mathbf{b})$ during the LM's forward pass. Now, consider a counterfactual input $\mathbf{s}$ which produces a counterfactual representation $f(\mathbf{s})$ when processed by $f$. We want to understand what about the output of $p$ is dependent on $f$. Therefore, we perform an intervention which replaces the output $f(\mathbf{b})$ with that of $f(\mathbf{s})$ during the computation of $p(\mathbf{b})$, with the change propagating downstream. The result is notated $p_{f \leftarrow f^*}(\mathbf{b}, \mathbf{s})$.

**Concrete setup for AR and ATR.** We take $\mathbf{o}$ to be a ground-truth document from our data distribution and $\mathbf{c}$ to be a version of that document with exactly one important token corrupted: the *key* (see Figure 2). This corruption significantly reduces task accuracy by removing information that is necessary to answer the AR query.

For each architecture, we intervene at both the input and output each of the following model components $f$: each layer block, each sequence-mixer (e.g. Attention blocks in each Transformer

| | Original | | | | | Corrupted Key | | | | | Restored @ Key | | | | | Restored @ Value | | | | | Restored @ Query | | | | |
|---|---|---|---|---|---|---|---|---|---|---|---|---|---|---|---|---|---|---|---|---|---|---|---|---|---|
| Mamba | 6.5 | 74.4 | 79.4 | 88.7 | 90.3 | 0.6 | 3.2 | 5.5 | 4.2 | 4.9 | 1.0 | 70.9 | 5.8 | 3.1 | 5.3 | 1.4 | 3.1 | 4.6 | 4.5 | 5.4 | 6.5 | 3.4 | 79.4 | 88.7 | 90.3 |
| Hyena | 0.1 | 0.1 | 2.6 | 12.7 | 29.9 | 0.1 | 0.1 | 0.7 | 1.5 | 1.5 | 0.1 | 0.1 | 0.6 | 1.6 | 0.9 | 0.1 | 0.1 | 0.4 | 1.3 | 1.6 | 0.1 | 0.1 | 2.6 | 12.7 | 29.9 |
| H3 | 0.1 | 0.2 | 1.3 | 1.8 | 1.0 | 0.1 | 0.1 | 0.5 | 1.0 | 0.6 | 0.1 | 0.2 | 0.4 | 1.8 | 0.8 | 0.1 | 0.1 | 0.2 | 1.1 | 0.7 | 0.1 | 0.1 | 1.3 | 0.6 | 1.0 |
| DeltaNet | 1.1 | 66.7 | 85.3 | 55.5 | 84.8 | 0.1 | 0.8 | 1.2 | 0.6 | 1.3 | 1.0 | 0.8 | 1.2 | 0.5 | 1.1 | 0.1 | 1.1 | 1.6 | 0.9 | 0.5 | 0.1 | 66.7 | 85.3 | 55.5 | 84.7 |
| Based | 21.8 | 92.0 | 98.7 | 98.9 | 98.5 | 0.3 | 2.7 | 4.1 | 6.4 | 7.4 | 0.1 | 3.5 | 5.0 | 5.8 | 8.3 | 21.9 | 91.9 | 98.7 | 98.9 | 98.5 | 0.0 | 2.7 | 5.7 | 8.2 | 7.1 |
| BaseConv | 0.0 | 0.7 | 3.5 | 5.8 | 8.5 | 0.0 | 0.5 | 3.1 | 6.4 | 7.8 | 0.0 | 0.1 | 3.3 | 5.8 | 8.1 | 0.0 | 0.6 | 3.6 | 6.0 | 8.3 | 0.0 | 0.5 | 3.5 | 6.3 | 8.2 |
| Attention | 1.9 | 100.0 | 99.9 | 100.0 | 100.0 | 1.7 | 5.8 | 5.5 | 6.0 | 4.8 | 1.7 | 6.0 | 4.3 | 5.3 | 5.3 | 2.1 | 100.0 | 99.9 | 100.0 | 100.0 | 1.7 | 5.0 | 6.4 | 5.7 | 4.3 |
| | 16 | 32 | 64 | 128 | 256 | 16 | 32 | 64 | 128 | 256 | 16 | 32 | 64 | 128 | 256 | 16 | 32 | 64 | 128 | 256 | 16 | 32 | 64 | 128 | 256 |

Model dimension

Figure 3: **Associative recall**: Likelihood of correct answer without any interventions, after corrupting the key, and after restoring representations at the layer 1 block input with interchange intervention, on AR with vocabulary size 8192 and key–value count of 32. SSMs (except for Based) and Transformers learn different mechanisms.

layer), and each state-mixer (an MLP, except in Mamba, which lacks this component).[2] We measure to what extent the intervention can restore the likelihood of the correct answer to the query, i.e. we compare **restored likelihood** $p_{f \leftarrow f^*}(y_{\text{true}} \mid \mathbf{c}, \mathbf{o})$ with original likelihood $p(y_{\text{true}} \mid \mathbf{o})$ and corrupted likelihood $p(y_{\text{true}} \mid \mathbf{c})$.

## 4 UNDERSTANDING AR WITH MECHANISTIC METRICS

We now deploy our mechanistic metrics (§3) on AR. We follow the methodology outlined in §4.1 to create a variety of datasets and train models with various architectures and hyperparameter configurations. See appendix D for additional experiments not included here.

### 4.1 METHODOLOGY

**Datasets.** We generate synthetic pretraining and evaluation datasets for AR. For each setting, the trainset has $100,032$ examples and the eval/dev sets have 320 examples. In AR, we use disjoint key and value vocabularies. In each document, we separate the document from the query with a divider token, and provide only a single query. Further details are in appendix C.

**Models.** We pretrain small models from scratch. We use the exact architecture implementations from the zoology[3] library (Arora et al., 2024b) as well as the DeltaNet implementation from fla[4] (Yang & Zhang, 2024), except for behaviour-preserving modification of the LM backbone to enable interventions with pyvene[5] (Wu et al., 2024) on various model-internal components. The LM backbone for all architectures is the same, with pre-norm blocks of alternating sequence mixers and MLPs (except for Mamba, which has no MLP) followed by LayerNorm at the end. We experiment with the following architectures: Attention (Vaswani et al., 2017), BaseConv (Arora et al., 2024a), Based (Arora et al., 2024b), DeltaNet (Yang et al., 2024), H3 (Fu et al., 2023), Hyena (Poli et al., 2023), and Mamba (Gu & Dao, 2024); further details are given in appendix A.

**Training.** We minimise next-token prediction cross-entropy loss, and mask the loss on all tokens except the *query*. We use the AdamW optimiser with $\beta = (0.9, 0.999), \epsilon = 10^{-8}$ and no weight decay. We warm up the learning rate for the first $10\%$ of training and then follow a cosine decay schedule to 0 for the remainder of training. We train for 16 epochs with a batch size of 32.

Each experiment trains $\approx 200$ models over all hyperparameters. Runtime varies from 0.5 to 5 hours, depending on hardware, task, and architecture. Overall, we used $< 10,000$ GPU-hours in total, on a cluster with various NVIDIA machines (GPU memory 12.3G to 143.8G).

---

[2]We are not limited to analysing only these components, however; we do experiment with mixer-internal representations in our analysis.

[3]https://github.com/HazyResearch/zoology

[4]https://github.com/fla-org/flash-linear-attention

[5]https://github.com/stanfordnlp/pyvene

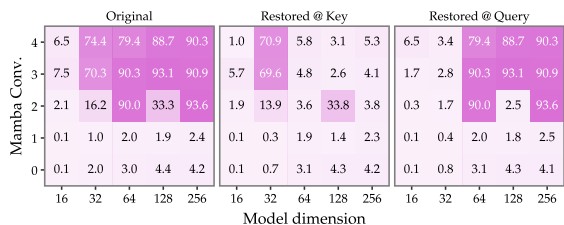 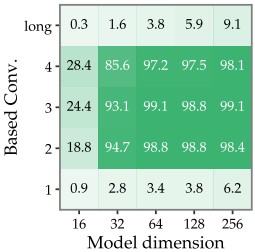

(a) Original and intervention restoration scores for **Mamba** when varying conv. kernel size; Restoring at the *value* is not shown since none of the models benefited from it.

(b) Accuracy on AR for **Based** when varying conv. kernel size or using implicit long conv.

Figure 4: **Ablating short convolution in Mamba and Based**: Accuracy and interchange intervention results when ablating parameters of the short convolution component in Mamba and Based. Conv. size less than 2, using long conv., or no conv., all lead to near-zero performance in both models.

**Behavioural metrics.** We report behavioural metrics given the model's predicted probabilities over the vocabulary $\hat{\mathbf{y}} \in \mathbb{R}^{|\Sigma|}$ and the index of the single true answer $i$. Our main metric is likelihood of the correct answer: $\hat{\mathbf{y}}_i$. We additionally report accuracy in appendices: $\mathbb{1}[\arg\max(\hat{\mathbf{y}}) = i]$.

### 4.2 (MOST) SSMS DO NOT SOLVE AR WITH INDUCTION

We first run experiments on a standard AR task and show that interchange interventions empirically confirm architectural differences in how AR is solved. We fix the total number of unique keys and values in the vocabulary to be $8192$, and present $32$ key–value pairs in context. As noted above, our trainset includes $100,032$ examples. We vary model dimensionality in $\{16, 32, 64, 128, 256\}$ and sweep LR in the range $[3 \cdot 10^{-5}, 3 \cdot 10^{-2}]$ for each architecture.

**Behavioural results.** The leftmost panel of Figure 3 demonstrates that answer likelihood on AR cleanly separates Attention, which achieves $100\%$ likelihood and accuracy at $d \geq 32$, from nearly all SSMs. Based solves AR near-perfectly with roughly the same dimension-wise scaling curve as Attention, achieving a maximum likelihood of $98.9\%$ (accuracy of $99.06\%$). However, Mamba and DeltaNet are close behind and clearly better than other SSMs at AR, albeit achieving a less-than-perfect $90.3\%$ likelihood ($91.25\%$ accuracy) at $d = 256$.

**Mechanistic analysis.** After corrupting the *key*, we restore the original representation at the layer 1 block input at each of various positions (*key*, *value*, and *query*) and observe the resulting likelihood of the true answer. High restored likelihood at the *value* token causally indicates that the model is performing **induction**, whereas *query* indicates **direct retrieval** by the layer 0 block and *key* indicates the layer 1 block, respectively. We report results on the LR-tuned checkpoints of each architecture at each model dimension.

Our results in Figure 3 cleanly separate Attention and Based, which only perform induction, from other SSMs, which either perform direct retrieval (Mamba, DeltaNet, Hyena) or fail to learn the task (H3, BaseConv). SSMs perform direct retrieval at varying layers: the best-performing SSMs almost entirely perform direct retrieval at layer 0 via the *query* token, with only one Mamba checkpoint ($d = 32$) delays retrieval to layer 1. Jelassi et al. (2024) show that direct retrieval in SSMs has asymptotically worse capacity than the induction solution, and this is reflected in performance on AR.

### 4.3 SHORT CONVOLUTIONS ENABLE AR IN MAMBA AND BASED

We have observed that Attention, Based, and Mamba are the highest-performing architectures on AR. However, their underlying mechanisms differ: Attention and Based learn **induction**, a 2-layer mechanism which stores *key*–*value* associations at the *value* token as an intermediate step, whereas Mamba uses **direct retrieval**, a 1-layer mechanism which writes the association to the *query* token.

Importantly, Based and Mamba share a key architectural component: **short convolutions**. Based is a hybrid model with alternating short convolution and linear attention layers, while Mamba

Original

| LR | 16 | 32 | 64 | 128 | 256 |
|---|---|---|---|---|---|
| 3.e-02 | 0.5 | | | | 0.0 |
| 1.e-02 | 4.2 | 75.4 | 80.6 | 85.7 | 0.0 |
| 3.e-03 | 1.8 | 42.4 | 70.9 | 87.5 | 89.0 |
| 1.e-03 | 0.1 | 10.5 | 53.2 | 57.2 | 92.8 |
| 3.e-04 | 0.0 | 0.1 | 1.2 | 6.9 | 3.8 |
| 1.e-04 | 0.0 | 0.0 | 0.0 | 0.2 | 0.1 |
| 3.e-05 | 0.0 | 0.0 | 0.0 | 0.0 | 0.0 |

Corrupted

| LR | 16 | 32 | 64 | 128 | 256 |
|---|---|---|---|---|---|
| 3.e-02 | 0.3 | | | | 0.0 |
| 1.e-02 | 0.7 | 2.9 | 4.7 | 5.4 | 0.0 |
| 3.e-03 | 0.1 | 3.0 | 6.1 | 2.2 | 6.9 |
| 1.e-03 | 0.1 | 1.2 | 4.2 | 3.5 | 7.2 |
| 3.e-04 | 0.0 | 0.1 | 1.0 | 1.9 | 1.6 |
| 1.e-04 | 0.0 | 0.0 | 0.0 | 0.2 | 0.1 |
| 3.e-05 | 0.0 | 0.0 | 0.0 | 0.0 | 0.0 |

Restored @ 0, Value

| LR | 16 | 32 | 64 | 128 | 256 |
|---|---|---|---|---|---|
| 3.e-02 | 0.3 | | | | 0.0 |
| 1.e-02 | 0.2 | 4.4 | 3.0 | 4.4 | 0.0 |
| 3.e-03 | 0.2 | 4.1 | 6.3 | 2.7 | 16.6 |
| 1.e-03 | 0.1 | 0.7 | 2.3 | 3.9 | 5.9 |
| 3.e-04 | 0.0 | 0.1 | 1.4 | 2.0 | 2.2 |
| 1.e-04 | 0.0 | 0.0 | 0.0 | 0.2 | 0.1 |
| 3.e-05 | 0.0 | 0.0 | 0.0 | 0.0 | 0.0 |

Restored @ 0, Next Key

| LR | 16 | 32 | 64 | 128 | 256 |
|---|---|---|---|---|---|
| 3.e-02 | 0.5 | | | | 0.0 |
| 1.e-02 | 4.2 | 3.2 | 80.1 | 86.0 | 0.0 |
| 3.e-03 | 1.5 | 23.0 | 70.7 | 87.2 | 72.1 |
| 1.e-03 | 0.1 | 0.6 | 44.7 | 56.4 | 92.6 |
| 3.e-04 | 0.0 | 0.1 | 1.0 | 0.9 | 2.8 |
| 1.e-04 | 0.0 | 0.0 | 0.0 | 0.2 | 0.1 |
| 3.e-05 | 0.0 | 0.0 | 0.0 | 0.0 | 0.0 |

Restored @ 1, Value

| LR | 16 | 32 | 64 | 128 | 256 |
|---|---|---|---|---|---|
| 3.e-02 | 0.2 | | | | 0.0 |
| 1.e-02 | 1.2 | 71.5 | 6.3 | 6.3 | 0.0 |
| 3.e-03 | 0.1 | 4.8 | 5.8 | 1.7 | 6.1 |
| 1.e-03 | 0.1 | 1.5 | 6.0 | 3.1 | 8.5 |
| 3.e-04 | 0.0 | 0.1 | 1.1 | 1.2 | 1.1 |
| 1.e-04 | 0.0 | 0.0 | 0.0 | 0.2 | 0.1 |
| 3.e-05 | 0.0 | 0.0 | 0.0 | 0.0 | 0.0 |

Model dimension

Figure 5: **Interventions on short convolutions in Mamba**: Likelihood of the correct answer for all Mamba checkpoints (varying model dimension and LR) on AR; (from left to right) original performance, after corrupting the *key*, restoring at the layer 0 short conv output at the *value*, same but at the next *key*, and same but at the layer 1 *value* token. (Some runs failed due to high LR.)

applies a short convolution to the layer input before each SSM block. We hypothesise that this architectural component is necessary[6] for learning AR when using a subquadratic sequence mixer. We conduct experiments on AR where we shorten the convolution kernel size in Mamba (from the default $d_{\text{conv}} = 4$ to $\{3, 2, 1\}$, and deleting it) and replace the Based short convolution with implicitly-parametrised long convolution (Poli et al., 2023).

**Results.** We report results of our ablations in Figure 4. On Mamba (Figure 4a), we find a step change in task accuracy when increasing $d_{\text{conv}}$ from 1 to 2, which introduces previous token information and thus enables AR. Without short convolution, Mamba fails to learn AR. Also, no Mamba checkpoints store the *key* at the *value* (i.e. induction), so direct retrieval persists. Finally, besides $d_{\text{conv}} < 2$ like Mamba, implicit long convolution in Based also significantly harms AR performance (Figure 4b). We thus conclude that short convolutions are necessary for performing AR in Mamba and Based.

## 4.4 Actually, Mamba *does* do induction

We have established that *(a)* Attention and Based perform AR via induction, whereas Mamba and other SSMs use direct retrieval, and *(b)* Mamba and Based require short convolutions to succeed at AR. However, we have not elucidated what 'direct retrieval' is; its only apparent distinction from induction is that it is implemented using a single layer, but what is happening in that layer is unclear.

We have already observed in Based that the association step of AR can be implemented with a short convolution, and then retrieval can be handled by a component with a longer receptive field (linear attention). But §4.2 shows that this results in an induction mechanism in Based: short convolution, just like Attention, performs association by moving information about the key to its associated value. Might short convolution in Mamba serve the same role?

**Methodology.** We use mechanistic metrics to analyse each layer's short convolution components in all of our Mamba checkpoints (sweeping model dimension and LR). To enable Mamba-internal interventions, we use a native PyTorch implementation of Mamba and load in the weights from our hardware-optimised training runs; we confirm that unintervened performance is unchanged. As in §4.2, we corrupt the *key*, and attempt to restore *key* information by intervening at the hidden states outputted by the short convolution component in each layer. We report the likelihood of the correct answer after intervention, averaged over 64 inputs.

**Results.** Our results in Figure 5 show that short convolution does move the *key* information; however, only a single checkpoint ($d = 32$, LR $= 10^{-2}$) actually moves *key* information to the *value* (as expected in induction, cf. Attention), and that too in layer 1; instead, we observe in the remaining 9 checkpoints with non-negligible performance, the information is moved to the ***next*** *key* by the layer 0 short convolution. While moving both the key and value to the following key is unusual, it still satisfies our two-stage definition of induction, and thus we claim that Mamba too performs induction.

---

[6]Since Hyena also has a short convolution, this may not be *sufficient* for good performance on association.

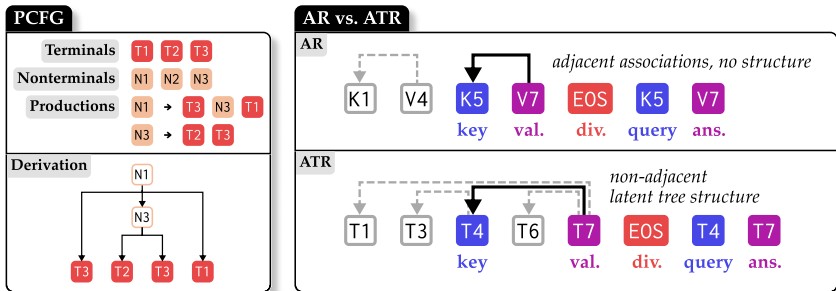

Figure 6: **PCFG**: An illustrative example of a PCFG and its components, with an example derivation (with final string) below. **AR vs. ATR**: Comparing AR and ATR using example documents; both tasks provide a document with key–value associations in-context and ask a query about one such association. However, associations in ATR need not involve adjacent tokens and are tree-structured.

## 5 MECHANISTIC FINDINGS ON AR GENERALISE TO A NEW RETRIEVAL TASK

We now examine whether our results generalise to a novel retrieval task: **Associative Treecall** (ATR). ATR is more similar to real-world natural language retrieval than AR because it uses a probabilistic context-free grammar (PCFG) to generate hierarchical data, on which we ask AR-like queries. Since *key* and *value* need not be adjacent to each other, ATR requires a non-positional associative mechanism, which may challenge architectures that are designed for AR.

### 5.1 ASSOCIATIVE TREECALL (ATR)

Since a standard AR document (eq. (1)) consists of *adjacent key–value* pairs, one can associate each *key* with its corresponding *value* solely using relative position. Yet many natural language retrieval tasks require association over latent hierarchical structure. For example:

$$(2) \qquad \textit{\underline{John} had \underline{chicken} and Mary had pork. The \underline{chicken} was eaten by} \rightarrow \underline{\textit{John}}$$

Answering this query requires associating *John* with *chicken* and *Mary* with *pork*, and then retrieving the appropriate association for *John*. A solution employing relative positional association would not robust to the possible range of variation (*John had some chicken*, *John decided to have chicken*, etc.).

An ATR corpus is drawn from a synthetic probabilistic context-free grammar (PCFG) whose parameters we set. Each document consists of a string sampled from the PCFG, with latent structure made up of **parent–child** relations between symbols, followed by a divider token (EOS) and a query about one such relation. The PCFG has one special property which establishes the parent–child relationships: for the right-hand side of each production rule, the rightmost symbol is always a terminal, and is the *parent* of the symbols created by this production. We sample strings by selecting an iid nonterminal and recursively applying production rules according to the PCFG distribution. We show an example in Figure 6 and formalise definitions in appendix B. Since the number of tokens separating parents and their children may vary, ATR cannot be solved by a positional associative mechanism.

**Setup.** For each experiment, we generate a single PCFG to use across all models to ensure fair comparisons, with parameters in Table 2. We also reject any samples that have more than 1024 symbols, which only affects the sampling distribution for the most complex PCFGs we use. We follow §4.1 except we train for 32 epochs.

Each PCFG sample of length $n$ provides us with a set of $n − 1$ eligible parent–child queries (i.e. a tree with $n − 1$ edges). However, terminals may occur multiple times, so a query about a specific symbol may present ambiguity; thus, when presenting a query we consider it to *only* refer to the rightmost instance of that symbol. To minimise the ability to heuristically guess, we inversely weight parent–child pairs by the parent's child count when sampling queries.

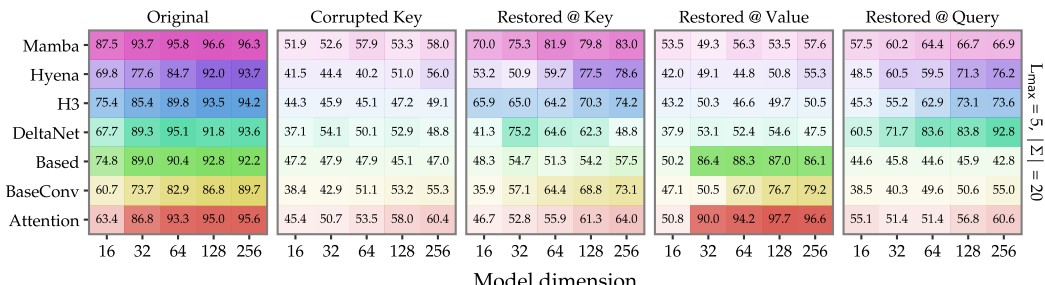

Figure 7: **Associative Treecall**: Likelihood of correct answer without any interventions (leftmost), after corrupting the key (second), and after restoring representations at the layer 1 block input (at *key*, *value*, or *query*) with interchange intervention. The pattern from AR largely holds.

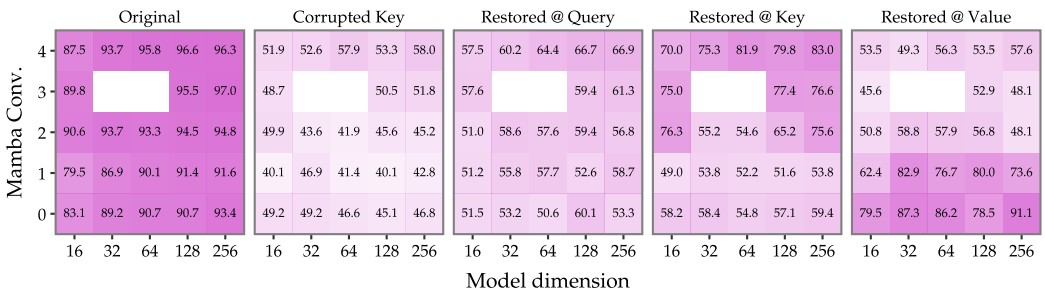

Figure 8: **Ablating short convolution in Mamba on ATR**: Original, corrupted *key*, and restored interventions at layer 1 input at the *query*, *key*, and *value* in Mamba, when varying model dimensionality and conv. size. (Some checkpoints failed to train.)

## 5.2 PER-ARCHITECTURE MECHANISMS ARE SIMILAR BETWEEN ATR AND AR

We consider a simple setting to study models on ATR: the maximum number of symbols on the right-hand side of any rule $L_{max} = 5$ and vocabulary size $|\Sigma| = 20$. The remaining parameters are set in accordance with appendix C; see Table 2 for definitions. We sweep the same model dimensionalities as in §4.2, and a smaller learning rate range of $[3 \cdot 10^{-5}, 3 \cdot 10^{-3}]$.

**Behavioural results.** We report results in the leftmost panel of Figure 7. Mamba is the best-performing architecture at ATR; it matches or outperforms all other architectures at all model dimensions. This is particularly surprising because longer production rules imply greater positional variation between keys and values, which ought to stress AR-optimised SSM designs like Mamba.

Unlike AR, the baseline performance on ATR after corrupting the key is well above 0. This is because some *key*–*value* pairings are more likely than others due to the underlying PCFG, enabling memorisation (unlike AR where all pairings are equally likely).

**Mechanistic analysis.** We conduct the same analysis as for AR. We recover the same overall trends but with greater inter-architecture variance: Figure 7 shows that Attention, Based, and BaseConv all primarily learn induction mechanisms with performance largely restored by *value* interventions, whereas the remaining SSMs perform direct retrieval as on AR, with performance being restored by intervening on the *key* (indicating direct retrieval by the layer 1 sequence mixer) or the *query* (indicating the same but by layer 0).

## 5.3 MAMBA CAN LEARN ATR WITHOUT SHORT CONVOLUTIONS

Given ATR and AR show similar patterns in mechanistic implementation across architectures, we now ask whether Mamba's reliance on short convolution holds on this new task. Since ATR may have longer and more variable distances between the *key* and *value* than AR, short convolution (with

a short receptive field that applies a fixed input-unaware transformation) may not be the appropriate operator for performing association.

**Methodology.** We repeat the experiments in §4.3 on ATR, training Mamba with varying conv. size or without conv. We apply mechanistic metrics to the best-performing checkpoints of each dimensionality and conv. setting.

**Results.** We report results in Figure 8. Unlike in AR, we find that Mamba still learns the task even without short convolutions (but not as well as with them). E.g. at $d = 256$, Mamba without short convolutions (93.4%) still outperforms Based (92.2%) and BaseConv (89.7%). The more surprising result is via mechanistic metrics: without short convolutions or with $d_{conv} = 1$, Mamba's performance after corruption can be restored primarily by intervening at the *value*, i.e. **Mamba without short convolutions learns two-layer induction on ATR**. We have thus shown that Mamba can indeed learn an induction-like mechanism to solve associative recall tasks, albeit using different components than Attention and only on certain tasks (ATR, but not AR). This may explain other induction-like mechanisms observed in Mamba trained on natural language, e.g. Bick et al. (2025), and highlights a divergence between AR and real-world retrieval.

## 6 DISCUSSION

**Why mechanistic evaluations over behavioural metrics?** Architectural advances on language modelling are largely uncovered and presented in an empirical manner (e.g. Shazeer, 2020); beyond intuition, we have little justification as to *why* a modification or innovation improves model performance. Synthetic tasks already inform progress on architecture design (such as SSMs), but treating such tasks as just a behavioural evaluation discards useful signal; control over task parameters presents an opportunity to explain performance using interpretability.

**ATR indicates induction is highly general.** We introduced ATR to break the naïve key–value adjacency of AR, and see whether general mechanisms underlying association still emerge across architectures. We find the same induction mechanism, where the association is computed and stored at the value before retrieval, in Attention and Based for both tasks, as well as in Mamba without convs. While Olsson et al. (2022) and later works define induction on adjacent tokens, ATR is evidence that a *position-independent* and generalising (appendix D.2) notion of association is worth studying.

**Short convolutions are key to association in SSMs.** We showed that Mamba and Based rely on short convolutions to learn how to associate keys and values on AR and ATR. Several earlier works point to the importance of short convolution: Arora et al. (2024b) empirically show its utility on AR, Allen-Zhu & Alfarano (2025) introduce a short convolution component (Canon) in various architectures, Olsson et al. (2022) show that 1-layer attention can learn induction if augmented with a length-2 convolution; also Liu et al. (2024b); Dolga et al. (2024); Fu et al. (2023); Poli et al. (2023).

**Limitations.** While we proposed mechanistic evaluations as a new tool, behavioural metrics like accuracy are still needed to properly contextualise results. Additionally, we focus on synthetic tasks throughout this work; extending our analyses to real-world models would help paint a more complete picture of the differences in capabilities and mechanisms of architectures on real-world tasks.

## 7 CONCLUSION

In this work, we introduce mechanistic evaluations as a powerful framework for comparing model architectures. This approach goes beyond high-level behavioural metrics, revealing substantive differences between architectures. Through analysis of synthetic in-context retrieval tasks, we uncover the underlying mechanisms that explain the success and failure points of various architectures. Mechanistic evaluations thus provide a useful tool for architecture design and analysis, as well as a new opportunity for interpretability research to open the blackbox of progress in AI.

## REPRODUCIBILITY STATEMENT

We provide our anonymised source code for reproducing all experiments at `https://anonymous.4open.science/r/tinylang-1061/`. We have specified the hyperparameters for training and evaluation as well as compute usage in appendix C and §4.1. We largely use open-source implementations of models from `zoology`[7] (Arora et al., 2024b) as well as the DeltaNet implementation from `fla`[8], with minor modifications for our mechanistic analysis with the open-source library pyvene[9] (Wu et al., 2024).

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

# Appendix

## Table of Contents

# A MODEL CONFIGURATIONS

Table 1: Default model configurations across all architectures. In experiments, we sweep learning rate and embedding dimension, reporting results from the instance with highest accuracy.

### (a) Attention

| Parameter | Values |
|-----------|--------|
| dropout | 0.0 |
| num_heads | 1 |

### (b) Hyena

| Parameter | Values |
|-----------|--------|
| l_max | 1024 |
| filter_order | 64 |
| num_heads | 1 |
| num_blocks | 1 |
| outer_mixing | False |
| dropout | 0.0 |
| filter_dropout | 0.0 |
| short_filter_order | 3 |
| bidirectional | False |

### (c) BaseConv

| Parameter | Values |
|-----------|--------|
| l_max | 1024 |
| kernel_size | $[3, -1]$ |
| implicit_long_conv | True |
| use_act | False |

### (d) Based

| Parameter | Values |
|-----------|--------|
| *BaseConv Layer* | |
| l_max | 1024 |
| kernel_size | 3 |
| implicit_long_conv | True |
| use_act | False |
| *Based Layer* | |
| l_max | 1024 |
| feature_dim | 8 |
| num_key_value_heads | 1 |
| num_heads | 1 |
| feature_name | taylor_exp |
| train_view | quadratic |

### (e) H3

| Parameter | Values |
|-----------|--------|
| l_max | 1024 |
| d_state | 1024 |
| head_dim | 1024 |

### (f) Mamba

| Parameter | Values |
|-----------|--------|
| d_conv | 4 |

| Param. | Description |
|---|---|
| $H$ | Is the head terminal at the left or the right of each production? |
| $d_{\max}$ | Maximum depth permitted for the PCFG to generate. |
| $L_{\max}$ | Maximum number of symbols of the right-hand side of a production rule. |
| $R_{\max}$ | Maximum number of production rules for each nonterminal. |
| $|\mathcal{N}|$ | Number of nonterminal symbols in the PCFG vocabulary. |
| $|\Sigma|$ | Number of terminal symbols in the PCFG vocabulary. |
| $r_\Sigma$ | Relative weightage on choosing a terminal when sampling production rules. |

Table 2: Parameters used for constructing a PCFG. We define PCFGs in Greibach Normal Form (GNF); see Appendix B for more details.

## B  FORMALISATION OF ATR

For reference, we provide formal definitions for PCFGs and the normal form we use in ATR.[10]

**Definition B.1.** A **probabilistic context-free grammar** is a tuple $\mathcal{G} = \langle \mathcal{N}, \Sigma, \mathrm{S}, \mathcal{R}, p \rangle$ where:

- $\mathcal{N}$ is a finite set of non-terminal symbols;
- $\Sigma$ is an alphabet of terminal symbols;
- $\mathrm{S} \in \mathcal{N}$ is a start symbol;
- $\mathcal{R} \subset \mathcal{N} \times (\mathcal{N} \cup \Sigma)^*$ is a finite set of production rules, mapping a left-hand side symbol $\mathrm{N} \in \mathcal{N}$ to a string of symbols that may be either terminals or nonterminals; each such rule is written as $\mathrm{X} \rightarrow \boldsymbol{\alpha}$;
- $p : \mathcal{R} \rightarrow [0, 1]$ is a weighting function which assigns a probability to each production rule for a nonterminal; this function is locally normalised, meaning $\{\sum_{\mathrm{X} \rightarrow \boldsymbol{\alpha}} p(\mathrm{X} \rightarrow \boldsymbol{\alpha}) = 1 \mid \mathrm{X} \in \mathcal{N}\}$.

**Definition B.2.** A PCFG $\mathcal{G} = \langle \mathcal{N}, \Sigma, \mathrm{S}, \mathcal{R}, p \rangle$ is in **Greibach normal form (GNF)** if each production rule in $\mathcal{R}$ is of the form $\mathrm{X} \rightarrow a\, \mathrm{X}_1\, \ldots\, \mathrm{X}_n$, where $\mathrm{X}_1, \ldots, \mathrm{X}_n \in \mathcal{N}$ and $n$ may be 0. Similarly, a PCFG is in **right-Greibach normal form** if each rule is of the form $\mathrm{X} \rightarrow \mathrm{X}_1\, \ldots\, \mathrm{X}_n\, a$.

For ATR, the PCFG is in Greibach normal form if the head is the leftmost symbol of the production rule's righthand side; similarly, if the PCFG is right-headed, it is in right-Greibach normal form.

**Definition B.3.** A **derivation step** $\boldsymbol{\alpha} \Rightarrow \boldsymbol{\beta}$ is an operation where, given strings of symbols $\boldsymbol{\alpha}, \boldsymbol{\beta} \in (\mathcal{N} \cup \Sigma)^*$, the leftmost nonterminal $\mathrm{X} \in \mathcal{N}$ in $\boldsymbol{\alpha}$ is rewritten using the right-hand side of a production rule $\mathrm{X} \rightarrow \ldots \in \mathcal{R}$ to obtain $\boldsymbol{\beta}$.

**Definition B.4.** A **derivation** under the PCFG $\mathcal{G}$ is a sequence of strings $[\boldsymbol{\alpha}_0, \ldots, \boldsymbol{\alpha}_m]$ where $\boldsymbol{\alpha}_0 \in \mathcal{N}$ and each step $\boldsymbol{\alpha}_{i+1}$ is formed by a derivation step on $\boldsymbol{\alpha}_i$. The final string $\boldsymbol{\alpha}_m \in \Sigma^*$ is the **yield** of the derivation.

Each ATR document is the yield of a derivation sampled under the GNF PCFG $\mathcal{G}$.

### B.1  ADDITIONAL DETAILS ON ATR

**Parent terminals in GNF.** We set the left/right-most terminal in each production rule (which leads to the GNF property) the parent of all other generated terminals. This terminal is sampled specially: for each nonterminal, we independently sample a distribution over terminals from a uniform Dirichlet, and for all production rules with that nonterminal on the lefthand side we use that distribution to sample the parent terminal. This simulates how heads of phrases in natural language (analogous to our parent terminals) decide the type of the phrase they head (analogous to our nonterminals).

**Maximum depth.** To enforce maximum depth, we first assign a uniformly random depth score $d : \mathcal{N} \rightarrow \mathbb{N} \in \{1, \ldots, \texttt{max\_depth}\}$ to each nonterminal in the vocabulary. Then, for each production rule for each nonterminal X, we only allow nonterminals Y with $d(\mathrm{Y}) > d(\mathrm{X})$ on the right-hand side. Note that this means no recursion is possible.

---

[10] We use similar formalisations of PCFGs as previous work in NLP, e.g. Nowak & Cotterell (2023).

# C  TASK HYPERPARAMETERS

Table 3: Task hyperparameters used for constructing key–value sets for AR and PCFGs for ATR. For a description of each parameter, see Table 2.

(a) Parameters used for constructing AR documents.

| Parameter | Values |
|-----------|--------|
| $L_{\max}$ | 32 |
| $L_{\min}$ | 32 |
| $|\Sigma|$ | {8192} |

(b) Parameters used for constructing ATR documents.

| Parameter | Values |
|-----------|--------|
| $H$ | Right |
| $d_{\max}$ | 10 |
| $L_{\max}$ | 5 |
| $R_{\max}$ | 5 |
| $|\mathcal{N}|$ | 40 |
| $|\Sigma|$ | 20 |
| $r_\Sigma$ | 20 |

# D   ADDITIONAL EXPERIMENTS ON AR AND ATR

Many parameters of synthetic tasks like AR and ATR and the model architectures we tested have interesting effects on behavioural and mechanistic metrics, but not all experiments could fit in our main text. Therefore, we include additional interesting observations in this appendix.

## D.1   ATTENTION NEEDS POSITION EMBEDDINGS

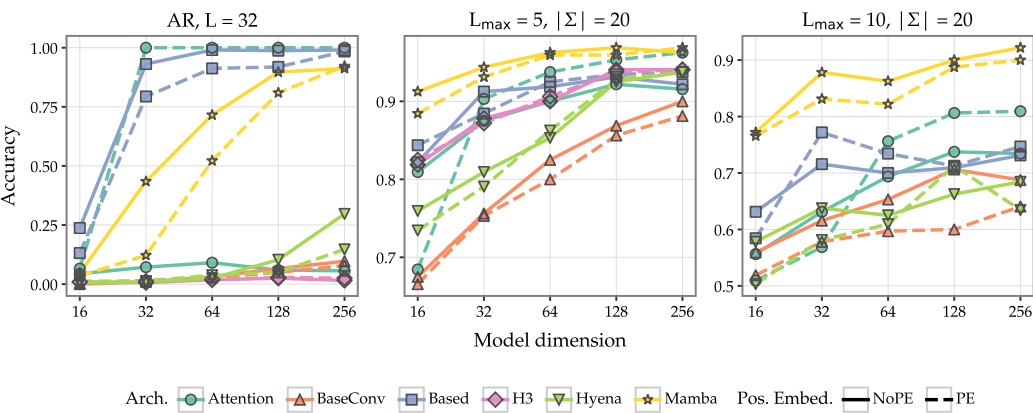

Figure 9: **Position embedding**: Model accuracy on AR and two ATR settings with and without absolute position embeddings.

Due to an initial configuration mistake, we accidentally trained all architectures with absolute position embeddings; in the `zoology` codebase (Arora et al., 2024a), only Attention is meant to be trained in this way. Fortuitously, this resulted in an interesting ablation: do SSMs, which are usually trained without it, also benefit from position embeddings?

**Behavioural results.** Our results in Figure 9 resoundingly show no: SSMs generally perform worse with position embeddings (PE). Attention is highly dependent on PE; performance on AR drops from $100.00\%$ to $5.62\%$ at $d = 256$ with NoPE. Attention lacks recurrence, unlike SSMs, so this is not surprising. However, on ATR, at smaller dimensionalities NoPE actually outperforms PE Attention. Further ablations ought to consider alternative PE methods such as RoPE and Alibi.

## D.2   MAMBA'S SOLUTION TO ATR DOES GENERALISE

We reuse the settings from our ATR experiment ($L = 5, |\Sigma| = 20$) and construct a new dataset with a train–test split on query–answer pairs. Specifically, $80\%$ of possible unique query–answer pairs are provided in the training set, while $20\%$ are only in the test set and thus never trained on. We seek to assess whether models learn a general mechanism for parent–child relations in ATR or if the impressive results of Mamba (as well as Attention and Based) are merely the result of better memorisation of the PCFG parameters. This setup is akin to Wang et al. (2024)'s technique of train–test split on multi-hop queries; we provide supervision on individual query and answer types, but not on some compositions of them.

**Behavioural results.** We select the checkpoint with the highest dev accuracy for each architectural and dimensionality setting, after sweeping LR. We plot the dev and test accuracies of each of these checkpoints in Figure 10a; all models have much lower test accuracy (e.g. Attention with $d = 256$ has $95.62\%$ dev and $68.12\%$ test accuracy). Attention achieves the greatest dev accuracies on $d \geq 32$. Mamba's relative ranking is lower than on the in-distribution setting in §5.2, but it still achieves the overall second-highest dev accuracy ($65.00\%$ at $d = 128$). Surprisingly, H3 generalises well despite its poor dev accuracy, beating Mamba on test accuracy in 3 out of 5 settings.

We compare dev and test accuracies across all LRs in Figure 10b. We find that while Mamba does have unusually high dev accuracy given a selected test accuracy (indicating greater memorisation than models with other architectures), its dev accuracy is still generally higher than non-Attention

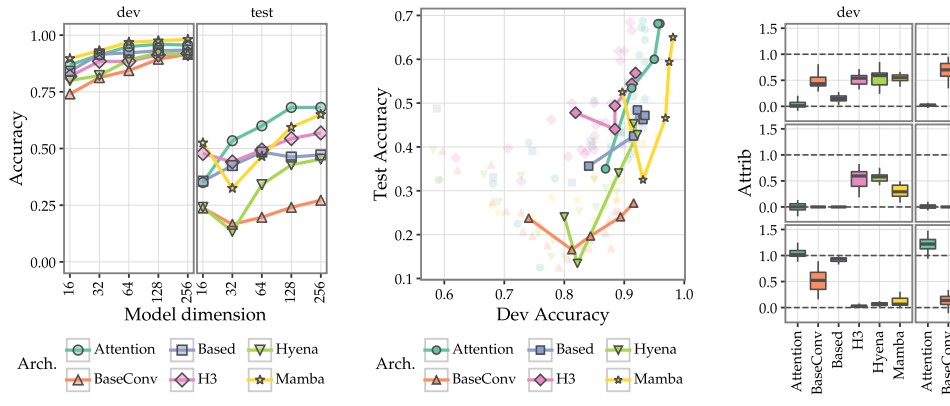

(a) Model dimensionality vs. accuracy on checkpoints with highest dev accuracy.

(b) Dev vs. test accuracy, with highest dev accuracy checkpoints at each dim. highlighted.

(c) Attribution scores for all checkpoints (except outliers), compared between dev and test sets.

Figure 10: **Generalisation on Associative Treecall**: Accuracy and interchange intervention results on ATR with train–test split. Scores are reported on dev (with in-distribution query–answer pairs from training) and test (OOD). We highlight the checkpoint with the best dev score in each setting.

architectures. Interestingly, H3 has nearly Attention-level generalisation while BaseConv exhibits vanishingly little generalisation. Overall, behavioural metrics show that Mamba does nontrivially generalise on ATR, albeit not as well as Attention.

**Mechanistic analysis.** We report a summary of attribution scores at different tokens (*key*, *query*, *value*), comparing on dev and test sets across all checkpoints in Figure 10c. We find largely consistent mechanisms underlying behaviour on both dev and test, and these match attribution scores on ATR without train–test split. The only exception is that BasedConv does induction on the dev set but not nearly as much on the test set; its induction mechanism is more brittle than Attention and Based.

Overall, the induction mechanism is not more general than the direct retrieval mechanism; both Attention and Mamba show greater generalisation than other architectures despite their entirely different solutions, and our mechanistic evaluations confirm that this solution is consistent across in-distribution and out-of-distribution queries.

### D.3   1-LAYER SSMS LEARN DIRECT RETRIEVAL ON AR AND ATR

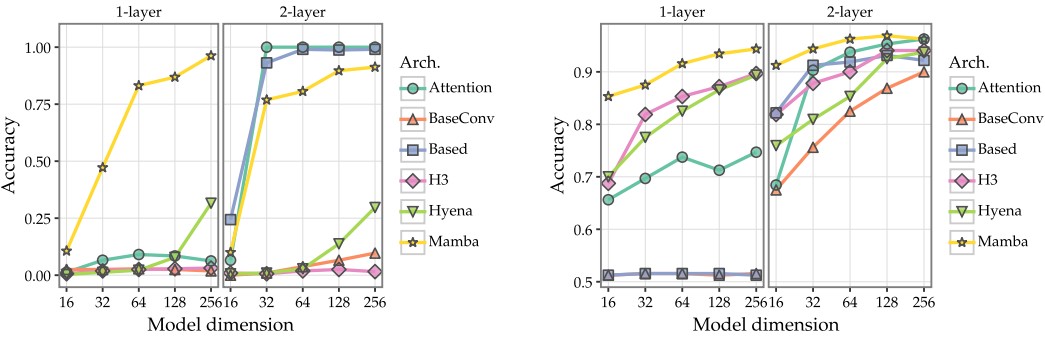

(a) Accuracy of 1-layer vs. 2-layer models on AR, 32 key–value pairs. 1-layer induction models fail.

(b) Accuracy of 1-layer vs. 2-layer models on ATR ($L = 5, |\Sigma| = 20$), with Based and BaseConv failing.

Figure 11: **1-layer models on AR and ATR**: Architectures that learn induction in the 2-layer setting fail to perform non-trivially with 1 layer. Mamba is highly performant with 1 layer on both tasks.

Throughout our experiments on AR and ATR, we have claimed that SSMs (except for Based and possibly BaseConv) learn a direct retrieval mechanism which does not require an intermediate step

like attention, i.e. only a single SSM layer is needed to learn AR and ATR. To verify this, we repeat AR and $L = 5, |\Sigma| = 20$ ATR experiments (without train–test split) with 1-layer models.

**Behavioural results.** We find comparable performance for direct retrieval models between 1-layer and 2-layer settings on AR (Figure 11a). In fact, at $d = 256$, 1-layer Mamba ($96.25\%$) outperforms 2-layer Mamba ($91.25\%$), as does Hyena ($31.56\%$ vs. $29.69\%$). 1-layer Based and BaseConv are architecturally identical, so we only report one; that architecture and Attention, both relying on induction in the 2-layer case, fail to learn AR with one layer. On ATR (Figure 11b), we see a more noticeable difference with layer count on all architectures, but again Attention, Based, and BaseConv become the worst architectures with one layer (e.g. $96.25\% \rightarrow 74.69\%$ for Attention at $d = 256$).

## D.4 SSMs PREFER LAYER 0 TO PERFORM AR

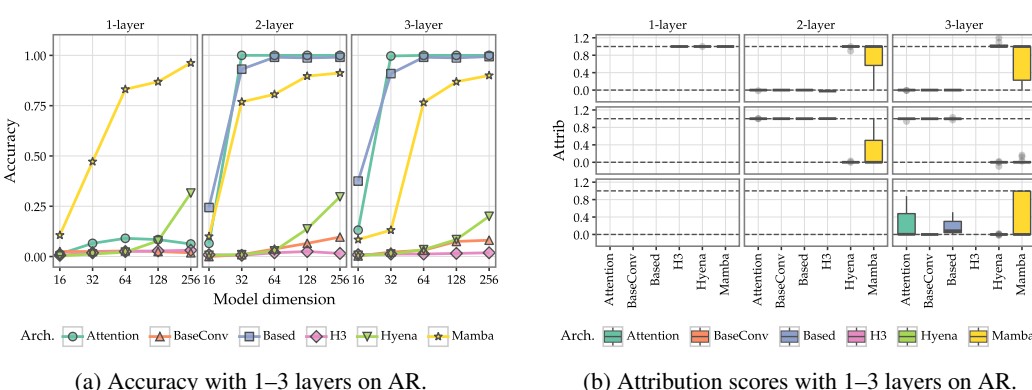

(a) Accuracy with 1–3 layers on AR.          (b) Attribution scores with 1–3 layers on AR.

Figure 12: **Varying layer count on AR**: Behavioural and mechanistic evaluations for models with 1–3 layers on AR.

Since we have confirmed that the direct retrieval mechanism in SSMs requires only a single layer, we are curious which layer this mechanism forms in if more than two layers are present. We train models with up to three layers on AR and report results.

**Behavioural results.** 3-layer models perform about the same on AR as 2-layer models across architectures (Figure 12a), except for a large drop in performance for Mamba when $d = 32$; this may just be an optimisation failure.

**Mechanistic analysis.** For our mechanistic metric, instead of intervening on each block, we intervene at the sequence mixer's output to the *query* token in each layer; this tells us if that layer is directly responsible for writing the answer to the output position. We apply the same filter as in §4.2, with a threshold of $0.01$. Figure 12b shows that among performant models, Hyena and Mamba prefer layer 0 for performing AR no matter the layer count; however, some Mamba checkpoints learn the mechanism in the final layer as well (but never layer 1 in a 3-layer model). Attention, Based, and BaseConv prefer layer 1, which is expected since this is the second step of the induction mechanism. However, some checkpoints of Attention and Based also have non-zero attribution score at layer 2 in the 3-layer setting.

## D.5 RIGHTMOST SIBLING QUERIES ARE TRIVIAL FOR ALL ARCHITECTURES

Since ATR has hierarchical structure, we attempted an initial experiment with multihop queries; specifically, we present queries where the answer is that terminal's rightmost sibling terminal. Models are only trained on this type of query, not standard parent queries as reported in the main text. We train with the same settings in §4.1.

**Behavioural results.** In Figure 13 we show that all models (except Based and BasedConv with 1 layer, where they only have local convolutions) achieve greater than $80\%$ accuracy at the task at all dimensionalities. We see slight improvement from 1-layer to 2-layer models but at this point performance is saturated and 3-layer does not help. Clearly, this task is extremely simple for all

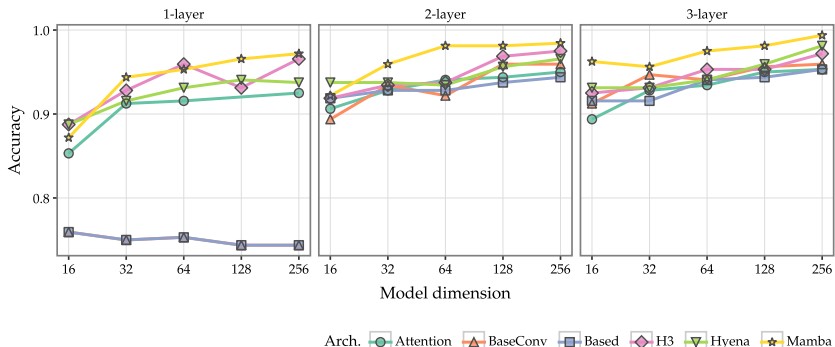

Figure 13: **Sibling queries**: Accuracy across models with 1–3 layers on ATR ($L_{\max} = 5, |\Sigma| = 20$)
.

models, even more so than parent queries, and thus does not provide useful signal for comparing architectures.

**Why are sibling queries easy?** Parent nodes are guaranteed to be special terminals in our GNF which are sampled from a nonterminal-dependent distribution (see appendix B). However, siblings have a large chance of being fixed terminals specified by the production rule. Additionally, the rightmost sibling of a particular terminal may be itself, if it is the rightmost terminal of its production rule. We speculate that these factors combined make sibling queries easier than parent queries, and thus not a suitable testbed for multihop reasoning.

**Future work.** The appropriate analogue to study multihop *reasoning* in ATR is grandparent relations (or higher up ancestors in the tree), since the grandparent is always a special head terminal (like the parent) and is always to the right of the parent and thus different from the query terminal. We leave further experiments on this to future work.

