# OpenReview forum: "Mechanistic evaluation of Transformers and state space models"
_ICLR.cc/2026/Conference — Submitted to ICLR 2026_

### Official Review · Reviewer_bLmn · 2025-10-30

**Soundness:** 2
**Presentation:** 3
**Contribution:** 2
**Rating:** 4
**Confidence:** 3

**Summary:**

The paper performs a mechanistic study on how SSMs and Transformers solve associative recall (AR) and associative treecall (ATR) tasks. The authors use causal interventions as the main tool. They find that for AR, Transformers and Based use induction mechanism, whereas SSMs such as Mamba rely on the convolution component --- without which Mamba fails to learn AR. They also find that similar mechanisms hold for ATR, except that Mamba can also use the Attention-like two-layer induction mechanisms in absence of the convolution component.

**Strengths:**

1. It is an interesting idea to use causal intervention as a mechanistic understanding tool to compare SSMs with transformers

2. The paper is clearly organized, with illustrative figures supporting the main claims.

**Weaknesses:**

1. Some claims are not well supported. For example, in line 75-76, the authors claim that ``(Mamba) fails to learn AR at all without this (short convolution component)'', but the experiments (Fig.4) only investigate the Mamba convolution kernel size no greater than $4$, unclear whether Mamba fails with long(er) convolution kernel.

2. Lack of architecture details and analysis. Although the authors provide source code, the paper lacks concrete definitions of the architectures used for study, and thus provides little explanation why certain mechanisms present in one architecture but not the others.

**Questions:**

1. Sec 4.4 Convolution moves information to the next key (line 317-322): This claim arises from the results in Fig.5, where restoring next key at layer 0 gives the best result. But why does the restoration necessarily imply the convolution component moves information to the next key? If my understanding of restoration is correct (which is not very precisely defined in paper, but loosely on Fig.2), restoration means we can arbitrary set any token at the next-key position. Then one easy solution is to set the next-key to be the corrupted key, which does not rely on convolution moving information to the next key. In addition, if the next key is arbitrary far away from the previous key, a short-convolution by definition cannot move information to the next key. Can the authors clarify?

 2. I appreciate the interesting use of causal intervention to mechanistically evaluate the AR mechanisms, but the findings seem to mostly corroborate the existing known results (e.g., short convolution are key to association in SSMs as mentioned by the authors in related work), including some provable solution mechanisms in missing related works [1] [2]. Does the mechanistic study offer novel insights (e.g., identify unknown mechanisms, provide more fine-grained analysis on how  the AR mechanisms interact with the choice of architecture and optimization set-up)?

 References:

 [1] Bietti et al. "Birth of a transformer: A memory viewpoint." NeurIPS 2023.

 [2] Huang et al. "Understanding Input Selectivity in Mamba: Impact on Approximation Power, Memorization, and Associative Recall Capacity." ICML 2025.

---

> ### Author Response · Authors · 2025-11-27
>
> Thank you for your review and the relevant questions you have raised. We will address your points below.
>
> **Weaknesses**
> 1. Our claim is that “Mamba does not learn AR without short convolution”, not “Mamba always learns AR with short convolution”. The latter, as you point out, is not proven to be true by our experiments, but we are not making that claim.
> 2. We use the exact implementations from `zoology` and `fla` libraries with hyperparameters given in the appendix. We can add additional details in the appendices for completeness.
>
> **Questions**
> 1. Restoration is formalised in section 3 and we believe it is important to understand this to assess our findings; we would like to know what parts were unclear so we can improve it, if you have the time to mention so in your response. To explain it briefly: Restoration occurs on hidden states, not input tokens; we take the representation from the original forward pass and overwrite the corrupted forward pass at some token position and layer in the model. If this intervention consistently changes the output to be correct, we have identified where the information used to solve the problem is being stored.
>     - Note that the tokens being considered are `[key] [value] [next key] … [query] [answer]`, where query == key and answer == value. The corrupted sequence is `[random key] [value] [next key] … [query] [answer]`, where it is not possible to obtain the correct answer since random key != query.
>     - Essentially, what we find in figure 5 is that if we patch the representation from the original sequence into the corrupted sequence, on the output of the short conv at the next key token position, we get the model to output the right answer. This means the information about [key] [value] is being moved by the short conv into [next key]. Since the window size is 4 this is possible; at [next key] the short conv operates on a window of 4 preceding tokens (including [next key] itself).
> 2. Yes this is true: prior work shows that convolutions are empirically good but **doesn’t explain or investigate why**. We state exactly this in line 466. As researchers, we desire scientific understanding of why certain decisions results in gains in performance. This paper aims to do exactly that and is quite up front about this goal. We believe our results are complementary to the progress in theory research, since we are able to directly intervene on models actually trained on the task and we don’t have to necessarily develop a theory for every SSM variant’s operators in order to make progress in understanding how the task is solved, under our methodology. We'd be happy to read and cite these works however since they are highly relevant.

---

### Official Review · Reviewer_LJok · 2025-10-30

**Soundness:** 2
**Presentation:** 2
**Contribution:** 2
**Rating:** 2
**Confidence:** 5

**Summary:**

The paper at hand presents a study of associative recall (AR) and associative tree recall (ATR) on various sequence models: from simple convolutions to Mamba and Transformers. The authors claim these models use different methods to solve AR, which they study by intervening on input sequences.

**Strengths:**

Studying basic performance tasks on transformers vs new sequence models is interesting. The authors train a large set of models and carry out the analysis also on ATR, which is a less common setup that I did not know before, but I find quite insightful.
The paper is also pleasant to read and schematic, which helps deliver the message. Plots are clear.

**Weaknesses:**

The findings in the paper have quite a few overlaps with previous works:

- Zoology: https://arxiv.org/abs/2312.04927
- Based: https://arxiv.org/abs/2402.18668
- Convolution-augmented transformers: https://arxiv.org/abs/2407.05591
- Revisiting Associative Recall: https://openreview.net/pdf/f7e9f322ba15e88dcc818ab70866648650a5e319.pdf
- H3 : https://arxiv.org/pdf/2212.14052

In light of the findings in the papers above, I did not find the paper very surprising. The authors cite all the papers above, but do not discuss or compare their results to previous literature:

1) performance on AR is reported in (1) "Zoology", and (2) with a much finer LR grid in "Revisiting Associative Recall". In the latter, the authors claim that LR sensitivity is a big issue in Mamba and Hyena. The authors do not discuss this issue nor seem to take any action to perform careful evaluations. Additionally, betas = (0.9, 0.999) in Adam is the default, but it is not what people typically use in language models. beta2 is too high (0.95 is default in many repos). How do you make sure your results depict what "each model can achieve"?

2) It is a bit hard to follow what the authors mean by "induction". I think, despite 1k years of philosophical debates, the definition can be a bit arbitrary. I was confused while reading your claims. Please define what you mean! I had an approximate idea at the end of the paper, but this is not formal enough.

3) The reader is not prompted to read the figures correctly, and the tasks are not well defined. Let us consider Figure 2: if you change A to B, the eval always returns A, and 2 returns "???". This is very unclear to me. I do not understand the ground truth and I find little explanations in the text. Furthermore, Figure 3 has a similar issue: you never formally define any of the tasks; what is "Restored @ Key"? You discuss how these tasks resemble interventions, but I cannot determine their respective importance. What are they individually supposed to test?

4) The role of convolution has been studied thoroughly, in "power of convolution-augmented transformers" but also in the H3 paper. In H3, they place a shift + gate exactly to enhance recall (https://arxiv.org/pdf/2212.14052, Fig 1). Again, I do not find surprising the claim about convolution, given also the induction head standard mechanism where the first transformer layer indeed represents a shift (e.g. proof in the Jelassi paper on the copy task, and Figure 1 above).

All in all, I do not see the level of novelty here to be at the level of acceptance. I ask the authors to please specify which new insights are presented in the paper, and to clarify what their causal interventions are precisely testing.

**Questions:**

See weaknesses.

---

> ### Author Response · Authors · 2025-11-27
>
> Thank you for your detailed review. We will do our best to respond to each point, and we particularly want to make sure our formalisation of causal interventions in section 3 is clear because it is necessary to understand that in order to assess our paper.
>
> 1. On LR and betas:
>     - We do sweep LR over a finer grid than Zoology; while we do state the range of the sweep for each experiment (e.g. line 242), we indeed missed reporting in the text the granularity of our range and will correct this. As figure 5 shows, we swept {3e-5, 1e-4, 3e-4, 1e-3, 3e-3, 1e-2, 3e-2}. Our accuracies for Mamba on AR match (2) “Revisiting Associative Recall” and this is by design, so our LR sweep is sufficiently fine-grained.
>     - Zoology (1) also uses default AdamW with betas of (0.9, 0.999): https://github.com/HazyResearch/zoology/blob/0bb96652b8c7cbb4a0263db68994ecfec3222115/zoology/train.py#L147.
> 2. “Induction” is the specific mechanism learned by the Transformer architecture (when it has at least two layers) to solve associative recall tasks; this term is not introduced by us and is defined in Olsson et al. (2022). It’s only related conceptually to the 1k year long usage of the word “induction” in logic, and means something rather specific in architecture research. We only use induction to refer to this mechanism, but we agree our generalisation of this term to other architectures besides the Transformer (e.g. in figure 1) can be unclear, since we haven’t formally defined it. Our results show that all architectures which succeed at AR use some serial mechanism to implement association and then recall but can use different operators for each step (e.g. attention in one layer followed by attention in another layer, short convolution followed by Mamba SSM in a single layer, etc.). Perhaps we can just describe the mechanism directly (e.g. “two-layer AR”, “one-layer AR”) rather than using “induction” since the terminology is unclear.
> 3. We clearly define what restored probabilities refer to in section 3. These are the result of corrupting the input and applying interchange interventions (not something that “resemble[s] interventions” -- we are unsure what is meant by this, interchange interventions are rigorously defined in Geiger et al. 2024 as cited in our paper) to the hidden states of the model. None of these plots show new tasks. They are the result of applying these interventions on the same task. This is the fundamental methodology underlying our proposed mechanistic metrics; the figures and results can hardly be understood without understanding this section. Overall, we are adapting and applying widely studied and established causal interpretability methods to the task of distinguishing SSM architectures.
> 4. Yes this is true: prior work shows that convolutions are empirically good but **doesn’t explain or investigate why**. We state exactly this in line 466. As researchers, we desire scientific understanding of why certain decisions result in gains in performance. This paper aims to do exactly that and is quite up front about this goal. Additionally, we’d like to pose the following questions to the reviewer, since we were quite surprised by these points ourselves:
>     - Did you expect convolutions to be storing keys and value at the next key in Mamba?
>     - Did you expect Mamba to succeed at ATR (a variant of AR) without short convolutions, but totally fail at AR with the same modification?
>     - Did you expect a variety of architectures (SSM, Based, Mamba) to implement the same mechanism with different operators (e.g. attention vs. short conv for association) to solve AR and a variant task?
>
> **References**
> - Olsson et al. (2022): https://arxiv.org/abs/2209.11895

---

### Official Review · Reviewer_wtMM · 2025-11-01

**Soundness:** 3
**Presentation:** 4
**Contribution:** 3
**Rating:** 6
**Confidence:** 3

**Summary:**

The paper introduces a generalized Associative Recall task named Associative Treecall and provide mechanic interpretability for the success of Mamba.

**Strengths:**

1. comprehensively and mechanically evaluate common linear models and transformer.
2. clear and good writing.

**Weaknesses:**

1. mechanic metric are not used to provide guidance for the design of architecture but can only help understanding. thus its use is limited.

**Questions:**

1. can you provide an example of how mechanic metric helps the design of architecture?

---

> ### Author Response · Authors · 2025-11-27
>
> Thank you for your review. Our current paper is centered around a case study showing that mechanistic analysis discriminates among a variety of SSM parameterisations and clearly favors the Based architecture for the critical (highly general) task of associative recall, so it does offer suggestions on choice of architecture.

---

### Official Review · Reviewer_UuRe · 2025-11-01

**Soundness:** 3
**Presentation:** 3
**Contribution:** 2
**Rating:** 4
**Confidence:** 4

**Summary:**

This paper introduces mechanistic evaluations (causal interchange interventions) to analyze the impact of different architectural components in solving Associative Recall (AR) task, along with a new retrieval task, called Associative Treecall (ATR). The authors show that on AR, Attention and Based architectures exhibit induction while Mamba and DeltaNet perform direct retrieval. The main ablation revolves around the size of the convolution kernels, where the ablations show that short convolution kernels are critical for AR on Mamba and Based architectures. With a detailed analysis on the two tasks, the authors claim that comparable accuracies can hide different internal mechanisms.

**Strengths:**

- The intervention protocol pinpoints the mechanism - more specifically, the QKV positioned restorations at layer input/outputs disambiguate induction from direct retrieval.
- The kernel size ablation study demonstrates that short receptive-field convolution kernels implement the association needed for AR.
- With the new task (ATR), the paper verifies that the mechanism transfer to a harder task without positional dependence.

**Weaknesses:**

- The experiment results seem quite noisy. For figure 4, in particular, it is unclear why model dim 128 and Mamba Conv. = 2 suddenly fails. Also, it is unclear why Restored @ Key value for this configuration is notably high compared to other entries. A similar trend is observed for Figure 5, where model configurations that perform well can drastically fail, (~0% accuracy) with learning rates that are slightly modified. This result, unless it can be justified empirically or theoretically, raises a serious concern whether the results are valid conclusions or are due to insufficient sweep in the hyperparameter space.
- Potential confounds with the parameter count on the models. The models have different parameter counts and FLOPs, and hence presenting the results in one of these dimensions would be much more valuable.

**Questions:**

- Please correct me if I misunderstood, but ATR induces unequal pair frequencies, which may induce a generative prior, allowing the corrupted-key accuracy to stay above 0 in most cases.
- Replacing Based's short convolution kernel with an implicit long convolution kernel significantly harms AR. Why might this be? Providing heatmaps for the long convolution kernel case to show where association fails could help better understand this phenomenon. Also a more detailed discussion about this phenomenon would strengthen this manuscript.
- The authors note that Hyena includes a short convolution kernel but it performs poorly on AR. This signals that a convolution kernel is not sufficient to solve this task of AR. Which downstream component could be preventing the convolution kernel from implementing the association step?

---

> ### Author Response · Authors · 2025-11-27
>
> Thank you for the detailed and useful response. We will do our best to address your points below.
>
> **Weaknesses**
> 1. It is empirically well-known that Mamba and other SSMs are far more sensitive to learning rate than Transformers, especially on the Associative Recall task: Okpekpe and Orvieto (2025) point this out and rectify prior results in the literature by sweeping over a finer learning rate grid, and their results also show that very small differences in LR can crater performance, especially in the high LR regime (~1e-2). Due to compute constraints we weren’t able to sweep quite as finely as them (our grid is half as dense) but we do sweep LRs more finely than other work, e.g. Arora et al. (2023). We thus argue that our results in this regard are not at all unusual in the context of prior work. We will add the results for figure 5 over all LRs we sweeped in the appendix of the paper.
> 2. We agree that the relationship between parameter count and FLOPs varies across architectures and would be useful to report. We follow prior work (e.g. Arora et al. (2023), sec. 4.1 of Yang et al. (2025)) in reporting results with respect to model width across architectures on AR, since the model width is a proxy for the in-context memory capacity of the model and this is the key metric of interest in AR, even when comparing across architecture. We can additionally report the FLOPs vs. performance results in an appendix, but do note that FLOPs is entirely determined by model width and architecture (since data size is held constant in our experiments).
>
> **Questions**
> 1. Yes this is true; unlike AR, ATR query-answer pairs are not uniformly distributed and so the model can memorise this prior to get seemingly above-chance performance. A version of ATR without this property would be more comparable to ATR, and we would like to come up with such a task in future work to better understand long-range association mechanisms. We will update the text to make this detail clear.
> 2. Re: Based implicit long convolution and Hyena short convolution, we absolutely agree these are interesting questions worth exploring, and we would really like to go beyond focusing on Mamba. In an earlier reviewing cycle, reviewers wanted a more extensive analysis of why convolution is important in Mamba and what kind of information it is responsible for propagating (similar to your question here); we updated the paper to address these points and thus obtained the surprising result that short convolution implements the association step, just like what happens in induction in Transformers. Unfortunately, we simply cannot fit every interesting experiment along these lines in a single paper, but we believe our primary contribution is that we introduce a method which the research community can apply to arbitrary mechanistic questions about architecture design. We hope this approach serves as a template for future research on understanding architectures.
>
> References
> - Okpekpe and Orvieto (2025): https://openreview.net/pdf?id=CcqAd5RPk5
> - Arora et al. (2023): https://arxiv.org/abs/2312.04927
> - Yang et al. (2025): https://arxiv.org/pdf/2406.06484

---

### Author Response · Authors · 2025-11-28

Thanks to all of the reviewers for raising important and thoughtful questions about our work. For the area chair (as well as all reviewers), we would like to summarise the main points of our rebuttals in order to make our position clear.

1. **Lack of understanding of/familiarity with causal interventions.** Broadly, we are concerned that reviewers did not understand the fundamental methodology we are using in this work. Our argument is that we should go beyond behavioural metrics to understand how different architectures learn simple tasks we care about -- using **causal interventions**, we can investigate the internals of these models to check if they learned the algorithm we want them to learn. We focus our study on associative recall (AR) since it is known to be fundamental to in-context understanding and can be studied in a very simple synthetic setting: the model is shown pairs of keys and values in context and then asked to retrieve the value corresponding to a particular key. We use existing techniques from the causal interpretability literature (see Geiger et al., 2025; sec. 2.1.3b of Sharkey et al., 2025; sec. 4 of Bereska and Gavves, 2024) to investigate how different LM architectures learn AR, and show the fundamental algorithm is similar across very different models, just implemented with different operators (e.g. association is done by attention in Transformers and by short convolution in Mamba).
    - Reviewer LJok in their point 3 reveal that **they do not understand how we are using causal interventions to reveal how information is flowing through the model** -- they ask what “tasks” we are studying, but in that section we only study one task (AR) and we explain how our interventions work in section 3 without using difficult notation or formalism. This is a surprising question given their confidence in their review is 5 -- while we are certainly applying a methodology that is not commonly used in LM architecture research, causal interventions are very widely used in areas such as interpretability and representation learning, and it isn’t unreasonable to expect reviewers to engage with this methodology.
    - Reviewer BLmn also **asks fundamental questions about causal interventions and may have missed our definition of restoration in sec. 3**.
2. **Justifying our hyperparameter sweeps.** Narrowly, reviewers (BLmn, LJok) asked questions about our training methodology, particularly hyperparameter sweeps. We fully deployed our limited compute resources to sweep learning rates and made sure to avoid pitfalls in studying the AR task (namely: not sweeping a fine-enough grid of LRs) that have been highlighted in prior work, e.g. Okpekpe and Orvieto (2025), Arora et al. (2023). We’ve also justified choices like the Adam betas and use of model width (instead of FLOPs) on this basis. In general, we think our methodological choices give a fair chance to every architecture we studied; some noise is expected.
3. **Why we prefer depth over breadth in our analyses.** Finally, some reviewers wanted more analysis of other interesting findings, e.g. why is Hyena underperforming Mamba despite also having a short convolution (from Reviewer UuRe)? Indeed, our analysis largely focus on Mamba, Based, and Transformers since those are the best-performing architectures. **We wanted to deeply understand a specific phenomenon: namely, why Mamba works so well on this task despite being quite different in architecture from Based and Transformers**. Since we are introducing a new methodology, providing a strong and detailed case study showing its utility is an important component of the paper, as opposed to providing many case studies which we may not be able to cover deeply due to page limits. We do want future work to analyse additional architectures but we don’t think our current paper would benefit from increasing breadth while sacrificing depth.

Thank you again for your engagement with our work.

**References**
- Geiger et al. (2025): https://www.jmlr.org/papers/v26/23-0058.html
- Sharkey et al. (2025): https://arxiv.org/abs/2501.16496
- Bereska and Gavves (2024): https://arxiv.org/abs/2404.14082

---

### Meta-Review · Area_Chair_YG9e · 2026-01-07

**Summary:**

This paper conducts a mechanistic evaluation of how various sequence modeling architectures (Transformers, Mamba, Based, H3, Hyena) solve the Associative Recall (AR) task. Using causal interchange interventions, the authors investigate the internal information flow of these models to determine how they recall information. The key findings are that while Transformers and Based utilize a two-layer "induction" mechanism (storing associations at the value token), Mamba relies heavily on its short convolution component to perform the association in a single layer. Specifically, the authors find that Mamba uses the short convolution to move key-value information to the next key, a distinct mechanism from Transformers. They further introduce a hierarchical task, Associative Treecall (ATR), showing that Mamba can learn induction-like behaviors on this task even when convolutions are removed.

The primary concerns are about the novelty and completeness of the mechanistic claims. While the application of causal interventions is methodologically sound, the core finding (that short convolutions are responsible for local association in SSMs) is largely confirmatory of existing empirical literature (e.g., Zoology, Based, H3) rather than a surprising new insight. Furthermore, there is a significant gap in the causal logic presented: the authors identify short convolutions as the mechanism for success in Mamba, yet acknowledge that the Hyena architecture also possesses short convolutions but fails at the task. This discrepancy suggests that the identified mechanism is necessary but not sufficient, and the paper declines to investigate this failure mode, leaving the causal explanation incomplete. Finally, while the authors defend the noise in their results as inherent to SSM training, the lack of stability combined with the "incremental" nature of the mechanistic insight limits the impact of the work.

Consequently, I recommend (weak) Rejection. The paper represents a rigorous scientific application of interpretability tools, but the insights derived do not sufficiently advance our understanding of architecture design beyond what is already intuited or empirically known in the field. In addition, the submission lacks a strong champion among the reviewers. The only positive score (Reviewer wtMM) was a low-confidence, superficial review that did not substantively engage with the paper's contributions. In contrast, the most critical review (Reviewer LJok) provided strong evidence that the findings are incremental. Without a compelling argument for why these specific mechanistic confirmations unlock new capabilities or design principles, the paper falls below the bar for acceptance at ICLR.

**Reviewer Concerns:**

### Addressed concerns:
* Methodological clarifications were requested by reviewers bLmn and LJok, who initially found the intervention protocol (restoration) and terminology ("induction" applied to SSMs) unclear. The authors provided adequate clarifications in the rebuttal, linking their methods to established causal interpretability literature.
* Reviewer UuRe raised concerns about the noise and instability in the results (e.g., sudden performance drops with small LR changes). The authors successfully argued that this sensitivity is an inherent property of Mamba/SSM training on AR tasks (citing Okpekpe & Orvieto), rather than a flaw in their specific setup.

### Outstanding & critical concerns:
* Reviewer LJok argued that the findings are incremental with limited novelty. This is the primary reason for the recommendation. As noted by the most critical reviewer, the finding that "short convolutions are responsible for local association" is largely confirmatory of existing empirical literature (e.g., Zoology, Based, H3). The paper does offer one specific mechanistic detail, that Mamba stores key-value associations at the next key token rather than at the value token as in Transformers, which is a novel observation. However, this is a narrow finding that does not substantially change our understanding of why these architectures succeed or fail, nor does it provide actionable guidance for architecture design. While the causal verification is methodologically sound, the broader result itself is not surprising to domain experts. The paper confirms that the component works as expected, rather than uncovering a novel or counter-intuitive mechanism.
* Reviewer UuRe raised concerns about incomplete causal explanation. The paper posits that short convolutions are the key to Mamba's success on AR. However, as noted by reviewers, the Hyena architecture also possesses short convolutions but fails at the task. The authors acknowledged this interest but did not provide an investigation due to space/scope. This leaves a gap in the causal logic: if short convs are the mechanism, why do they not suffice for Hyena?

**Reviewer Scores:**

* **Reviewer UuRe: Original score: 4. Estimated score: 4**. While the rebuttal explained the noise, the reviewer's concerns about the lack of insight into why similar architectures (Hyena) fail remain valid. The response was a defense, not a fix.
* **Reviewer wtMM: Original score: 6. Estimated score: 6**. This reviewer did not engage deeply; their score likely remains static but carries low weight.
* **Reviewer LJok: Original score: 2. Estimated score: 2**. The author's response addressed methodological clarity but did not fundamentally resolve the reviewer's main objection: that the paper confirms known intuitions without significant novel insight. This reviewer is an expert in the sub-field and their stance is unlikely to change.
* **Reviewer bLmn: Original score: 4. Estimated score: 4. The technical clarifications were helpful, but the reviewer's concerns about the limited scope of the ablation studies (only testing short kernels) prevents a shift to a clear accept.

---

### Decision · Program_Chairs · 2026-01-26

Reject